# *In vitro* immunoreactivity and *in vivo* neutralization of *Trimeresurus gracilis* venom with antivenoms targeting four pit viper species

**Po-Chun Chuang**[1,2], **Jia-Wei Chen**[2], **Yuen-Ying Chan**[3], **Tsz-Chun Tse**[4¤], **Yu-Wei Chiang**[5,6], **Tein-Shun Tsai**[3,4] *

1 Department of Computer Science and Engineering, National Sun Yat-sen University, Kaohsiung, Taiwan,
2 Department of Emergency Medicine, Kaohsiung Chang Gung Memorial Hospital, Taiwan, 3 Department of Biological Science and Technology, National Pingtung University of Science and Technology, Pingtung, Taiwan, 4 Institute of Wildlife Conservation, National Pingtung University of Science and Technology, Pingtung, Taiwan, 5 Department of Medical Research, Taipei Veterans General Hospital, Taipei, Taiwan, 6 Department of Biology and Anatomy, National Defense Medical Centre, Taipei, Taiwan

¤ Current address: Department of Biological Science and Technology, National Pingtung University of Science and Technology, Pingtung, Taiwan
* t43013@gmail.com, tstsai@mail.npust.edu.tw

**Data Availability Statement:** All relevant data are within the manuscript and its Supporting Information files.

## Abstract

Snakebite envenomation is a significant global health issue that requires specific antivenom treatments. In Taiwan, available antivenoms target a variety of snakes, but none specifically target *Trimeresurus gracilis*, an endemic and protected species found in the high mountain areas of Taiwan. This study evaluated the effectiveness of existing antivenoms against *T. gracilis* venom, focusing on a bivalent antivenom developed for *Trimeresurus stejnegeri* and *Protobothrops mucrosquamatu*s (TsPmAV), as well as monovalent antivenoms for *Deinagkistrodon acutus* (DaAV) and *Gloydius brevicaudus* (GbAV). Our research involved *in vivo* toxicity testing in mice and *in vitro* immunobinding experiments using (chaotropic) enzyme-linked immunosorbent assays, comparing venoms from four pit viper species (*T. gracilis*, *T. stejnegeri*, *P. mucrosquamatus*, and *D. acutus*) with three types of antivenoms. These findings indicate that TsPmAV partially neutralized *T. gracilis* venom, marginally surpassing the efficacy of DaAV. *In vitro* tests revealed that GbAV displayed higher binding capacities toward *T. gracilis* venom than TsPmAV or DaAV. Comparisons of electrophoretic profiles also reveal that *T. gracilis* venom has fewer snake venom C-type lectin like proteins than *D. acutus*, and has more P-I snake venom metalloproteases or fewer phospholipase $A_2$ than *G. brevicaudus*, *T. stejnegeri*, or *P. mucrosquamatus*. This study highlights the need for antivenoms that specifically target *T. gracilis*, as current treatments using TsPmAV show limited effectiveness in neutralizing local effects in patients. These findings provide crucial insights into clinical treatment protocols and contribute to the understanding of the evolutionary adaptation of snake venom, aiding in the development of more effective antivenoms for human health.

**Funding:** This work was supported by the Ministry of Science and Technology (grant numbers 109-2621-B-020-001, 110-2621-B-020-001 to TST) and the Kaohsiung Chang Gung Memorial Hospital and National Pingtung University of Science and Technology Joint Research Program (grant number CGMH-NPUST-2022-CORPG8M0111 to PCC and TST). The funders had no role in study design, data collection and analysis, decision to publish, or preparation of the manuscript.

**Competing interests:** The authors have declared that no competing interests exist.

## Author summary

*Trimeresurus gracilis* is a protected pit viper species endemic to the mid-to-high-altitude mountainous areas of Taiwan. Due to the lack of a serum specifically targeting *T. gracilis* envenomation, patients bitten by this species currently receive treatment by employing a bivalent antivenom developed against *T. stejnegeri* and *P. mucrosquamatus*, which is believed to potentially possess cross-neutralizing effects. However, the use of the bivalent antivenom has demonstrated limited effectiveness in neutralizing local effects in patients. In response, we conducted investigations into the *in vitro* immunological reactivity and *in vivo* neutralization of *T. gracilis* venom using antivenoms developed against four pit viper species. Our results indicate that the bivalent antivenom was not completely effective in neutralizing *T. gracilis* venom. Instead, the monovalent antivenom for *Gloydius brevicaudus* exhibited higher *in vitro* binding capacities toward *T. gracilis* venom compared with other antivenoms. These findings may be attributed to a closer phylogenetic relationship between *T. gracilis* and *G. brevicaudus*. Furthermore, these observations have the potential to contribute to the development of more effective antivenoms, offering valuable insights for the community focused on neglected tropical diseases.

## Introduction

Snakebite envenomation is a global concern, and its primary treatment involves antivenom administration [1]. In Taiwan, based on the species of the offending snake, four types of antivenoms are supplied by the Center for Disease Control (CDC) [2]. The available range includes an antivenom specific for *Daboia siamensis*, one for *Deinagkistrodon acutus*, a bivalent antivenom for *Bungarus multicinctus* and *Naja atra*, and another bivalent antivenom designed to neutralize the venoms of *Trimeresurus stejnegeri* and *Protobothrops mucrosquamatus*. However, in Taiwan, there are instances of snakebite envenomation, including those caused by *Ovophis makazayazaya* and *Trimeresurus gracilis*, for which no corresponding antivenom currently exists [3–5].

In Taiwan, majority (73.3%) of envenomation events were associated with *T. stejnegeri* and *P. mucrosquamatus*, with case-fatality rates of 0.09%, during 2002–2014 based on the use of snake antivenoms [6]. Among Asian countries, Taiwan reported low incidence and case-fatality rates of snakebite envenomation, which might be due to easy access to modern medicine, immediate availability of antivenoms in healthcare facilities, few adverse effects of antivenoms, and prompt surgical intervention [6]. Only 0.6% of the envenomation events were associated with *D. acutus* [6]. As of now, there have been only two case reports of envenomations by *O. makazayazaya* [3,4] and two case reports of envenomations by *T. gracilis* in Taiwan [5].

The main effects of *T. stejnegeri* and *P. mucrosquamatus* bites in Taiwan were tissue swelling, pain, and local ecchymosis. In *T. stejnegeri* envenomation, bullae formation and coagulation function are generally normal, and patients can be administered one to two vials of the bivalent antivenom developed against *T. stejnegeri* and *P. mucrosquamatus* (TsPmAV) initially. In *P. mucrosquamatus* envenomation, bullae or necrosis is present, but coagulopathy and thrombocytopenia are uncommon. Patients can be administered two to four vials of TsPmAV initially and the median total dose of the specific antivenom was 5.5 vials [5–9]. *D. acutus* envenomation is usually associated with hemorrhagic bullae or necrosis, prolonged PT/activated partial thromboplastin time, thrombocytopenia, and patients can be administered two to four vials of the monovalent antivenom against *D. acutus* (DaAV) initially [5,6,8,10,11].

In *O. makazayazaya* envenomation, patients may exhibit low fibrinogen and elevated D-dimer levels, and treatment may be initiated with one vial and completed with a total of five vials of TsPmAV [3,4].

*T. gracilis*, also known as Kikuchi habu, is found in mid-to-high-altitude mountainous regions of Taiwan. It is classified as a protected species, typically inhabiting the forest floor, near streams, and piles of rocks along forest trails [12]. Currently, patients bitten by *T. gracilis* are primarily treated with TsPmAV. However, the case reports available to date indicate that patients bitten on the finger by this species, not only experience local swelling, bruising, hemorrhagic blisters, or tissue necrosis but also systemic symptoms such as coagulopathy [5]. Moreover, even after immediate administration of 6–12 vials of TsPmAV within 20 h post-bite, patients still require surgical debridement or skin grafting [5]. This casts doubt on the effectiveness of TsPmAV for cross-neutralization.

Cross-neutralization tests using antivenoms and snake venoms are instrumental in determining which antidotes are effective in treating envenomation by specific snake species [13–16]. However, the effectiveness of antivenoms in neutralizing the venom of *T. gracilis* remains largely unexplored in specialized research studies. Although *T. gracilis* and *T. stejnegeri* were previously classified within the same genus, *Trimeresurus*, numerous recent studies on their phylogenetic relationships have indicated that they should not be grouped together [17–20]. Furthermore, recent research has revealed that the amino acid composition of some metalloproteinases, serine proteases, and phospholipase $A_2$ enzymes in the venom of *T. gracilis* is similar to those found in Asian pit vipers (*Gloydius*) and rattlesnakes (*Crotalus*) [21,22]. Therefore, the neutralizing potential of antivenoms produced for closely related snake species, such as *Gloydius brevicaudus* or rattlesnake species, against *T. gracilis* venom also requires further investigation and clarification.

In this study, we conducted *in vivo* toxicity tests on the venom of *T. gracilis* in mice to assess the degree of cross-neutralization by TsPmAV or DaAV. In addition, we compared the *in vitro* immunological binding capabilities of venom from four pit viper snakes (*T. gracilis*, *T. stejnegeri*, *P. mucrosquamatus*, *D. acutus*) with TsPmAV, DaAV, and the monovalent antivenom against *G. brevicaudus* (GbAV). We also performed electrophoretic analysis of *T. gracilis* venom, comparing it to venoms from other snake species, to augment the discussion on antivenom binding capacity levels. This study aimed to provide in-depth insight into the antivenomics of *T. gracilis* venom. Furthermore, this study sought to enhance our understanding of the evolutionary adaptation of this venom, the development of effective antivenoms, and to provide clearer evidence for clinical treatment post-envenomation.

## Methods

### Animal ethics

All animal procedures were approved by the Institutional Animal Care and Use Committee of the National Pingtung University of Science and Technology (approval no. NPUST-110-100, NPUST-112-010).

### Collection, husbandry, and venom extraction of *T. gracilis*

**Snake collection.** The collection sites for *T. gracilis* specimens included locations such as Taipingshan and Tataka. Venom was extracted from six snakes. Long rubber boots were worn for protection during field expeditions to locate and capture these snakes. A snake hook exceeding the effective striking range of the snakes was used to safely transfer the snakes into containers. This technique prevents the need for direct handling of snakes (especially in the

wild), thereby minimizing the snakebite risk. Subsequently, the containers were securely placed in transport bags to ensure that the snakes did not escape during transportation.

**Husbandry.** The snakes collected were placed in specialized herpetological trait recording boxes [23] to document their basic morphological data, such as snout-vent length and tail length (to the nearest 0.1 cm); body weight (to the nearest 0.1 g) was also measured. Additionally, the sex and reproductive status (e.g., pregnancy) of each snake were determined. The snakes were then housed in a herpetarium under controlled conditions: air temperature of 20–22˚C, relative humidity of 50–80%, and a 12L:12D light cycle, continuously monitored using instruments. Each animal was individually housed in a plastic enclosure measuring 638 mm × 425 mm × 166 mm, with a usable interior volume of 31 liters. The enclosure was equipped with ventilation holes in the lid and walls, along with a thermal gradient, a sphagnum substrate, hiding spots, and water dishes for animal comfort and well-being. This setup aimed to enhance environmental enrichment, allowing snakes freedom of movement and the ability to choose their preferred microenvironment.

**Venom extraction.** Venom extraction was scheduled in accordance with the standard operating procedures established for venomous snakes in animal facilities. All handlers wore long rubber boots and snakebite-proof gloves to ensure safety. Before venom extraction, the health of each snake was assessed to ensure the absence of visible diseases, oral abscesses, and gingivitis. Clean containers, pre-weighed and covered with a thin layer of parafilm or plastic film, were prepared for venom collection to facilitate snake biting and venom release. The snake intended for venom extraction was carefully removed from the enclosure using a snake hook. The handler secured the snake's head and neck with one hand to prevent it from turning and causing injury, while gently holding the rear or tail of the snake with the other hand. The snake was then induced to bite through a thin film covering the container, releasing venom. The venom from each snake was collected separately in sample tubes, and labeled, and the wet weight (including water content; to the nearest 0.01 mg) of the fresh, undried venom was recorded. Freeze drying was conducted in an FD-series freeze dryer and a CES-series centrifugal evaporator (Pan-chum Scientific Corp., Kaohsiung, Taiwan). After drying, the dry weight of the venom was measured (to the nearest 0.01 mg), and the samples were stored at -20˚C for subsequent biochemical experiments with the venom. Following venom extraction, the snakes were released back to their original collection sites.

## Chemicals and antivenoms

All the chemicals and reagents used were of analytical grade. Bovine serum albumin (BSA), Tween-20, ortho-phenylenediamine (OPD), anti-horse IgG F(ab')$_2$-peroxidase antibodies produced in goats, ammonium persulfate, sodium citrate tribasic dihydrate, and citric acid were purchased from Sigma-Aldrich (Burlington, MA, USA). Phosphate-buffered saline (PBS; pH 7.5) was obtained from UniRegion Bio-Tech (New Taipei, Taiwan). Protein assay dye, acrylamide/bis solution, 4x Laemmli sample buffer, tetramethylethylenediamine, and Coomassie Brilliant Blue R-250 were purchased from Bio-Rad (Hercules, CA, USA). The ExcelBand 3-color broad-range protein marker (5–245 kDa) was obtained from Smobio (Hsinchu, Taiwan). Tris hydrochloride, Tris Base, and sodium dodecyl sulfate were purchased from J.T. Baker (Center Valley, PA, USA). Methanol and acetic acid were purchased from Nihon Shiyaku Reagent (Kyoto, Japan). Hydrogen peroxide (H$_2$O$_2$) and hydrochloric acid (HCl) were purchased from Honeywell (Charlotte, NC, USA). Non-immunized horse IgG was purchased from Innovative Research (Novi, MI, USA). Two types of antivenoms, TsPmAV (batch number: 60-06-0017; expiry date: May 19th, 2025) and DaAV (batch number: 60-06-0014; expiry date: Jan 30th, 2025), were purchased from the Ministry of Health and Welfare Centers

for Disease Control (CDC), Taipei, Taiwan. These agents were freeze-dried F(ab$'$)$_2$ antibody preparations derived from horses, with a neutralizing potency of more than 1,000 Tanaka units [15,24]. Additionally, another type of F(ab$'$)$_2$ horse-derived antivenom serum, GbAV (contains 6000 IU/vial of 10 mL; batch number: 20200603, expiry date: Jun 12$^{th}$, 2023), was obtained from Shanghai Serum Bio-Technology (Shanghai, China). All antivenoms were used before expiry.

## Snake venom protein quantification

The individual venom samples from *T. gracilis* were pooled in equal proportions before analysis. Additionally, for comparisons, we included the respective pooled venom samples from other species (*T. stejnegeri*, *P. mucrosquamatus*, *D. acutus*) in the analysis. Subsequently, the pooled venom samples were dissolved in ultrapure water and centrifuged at $10,000 \times g$ and 4°C for 10 min. The protein concentration in the supernatant was determined in triplicate using the Bradford protein assay method [25]. In each well of a 96-well microtiter plate, 10 μL of protein standard (BSA) or snake venom sample was added. Subsequently, 200 μL of protein assay dye was added to each well, taking care to avoid bubble formation. The plate was gently tapped on one side to ensure a thorough mixing of the components. The plates were incubated at room temperature for 5 min. The optical density at 595 nm (OD$_{595}$) was measured using an ELISA plate reader (AMR-100, Medclub Scientific Co., Ltd., Taoyuan, Taiwan). A standard calibration curve was plotted to determine the protein concentration in the venom samples.

## Sodium dodecyl sulfate-polyacrylamide gel electrophoresis (SDS-PAGE)

The pooled venom samples (15 μg) from each species were dissolved in 4x Laemmli sample buffer and loaded to 12.5% SDS-PAGE gel under non-reducing and reducing conditions [26,27]. The ExcelBand 3-color broad-range protein marker (5–245 kDa) was used as a calibration standard. Electrophoresis was performed at 110–150 V for 2 h. The gel was stained with Coomassie Brilliant Blue R-250 and destained for visualization. Gel images were captured using the Universal Hood II Gel Doc XR System (Bio-Rad, CA, USA). Densitometric analysis for the protein bands in each lane of the SDS-PAGE gels were performed using the Image Lab 6.1 software (Bio-Rad, CA, USA). We measured the intensity of all bands within each venom sample and then calculated the relative abundance of each band based on its intensity.

## Determination of median lethal dose (LD$_{50}$) of *T. gracilis* venom

All male ICR/CD-1 mice used for toxicity tests were procured from BioLASCO, Taipei, Taiwan. Nine groups of male ICR/CD-1 mice (16–18 g, 5 mice per group) received intraperitoneal injections of 100 μL of *T. gracilis* venom. The venom was administered at doses of 0.4, 0.8, 1.6, 2.0, 2.4, 2.8, 3.2, 4.8, 6.4, and 9 μg/g. All doses were diluted in PBS. The mortality rate of the test animals was recorded 24 h post-injection at room temperature [28].

## Determination of neutralizing effects of antivenoms on *T. gracilis* venom lethality

To evaluate the neutralizing effects of the various antivenoms against the lethality of *T. gracilis* venom, the median effective dose (ED$_{50}$) was measured. A dose of snake venom, equivalent to three times the LD$_{50}$, was mixed with different dilutions of antivenoms and a control group (PBS buffer) at 37°C for 30 minutes. These mixtures were then administered intraperitoneally to 8–9 groups of male ICR/CD-1 mice, with each group comprising five mice. The mortality rate of the test animals was recorded at 24 h post-injection at room temperature.

### Analysis of antivenom binding capacity

**Enzyme-linked immunosorbent assay (ELISA).** In ELISA-specific 96-well plates, 10 ng of snake venom (dissolved in 100 μL PBS buffer) was added to each well in triplicate and left to adsorb overnight at 4˚C. After washing the wells five times with PBS containing 0.1% Tween-20 (PBST), 100 μL of 2% casein in PBS was added and incubated at room temperature for one hour to block unoccupied sites. The wells were washed five times with PBST and incubated with various dilutions of antivenom serum (1:100, 1:300, 1:900, 1:2700, 1:8100, 1:24300, and 1:72900) at room temperature for two hours to allow immunological reactivity. Non-immunized horse IgG preparations and PBS served as controls. After another five washes with PBST, 100 μL of anti-equine immunoglobulin-peroxidase conjugate diluted 1:50000 was added to each well and incubated for one hour at room temperature. The wells were then washed five times with PBST, followed by the addition of 100 μL of 2 mg/mL OPD (dissolved in 0.1 M sodium citrate buffer, pH 5.0 with 30% $H_2O_2$ at 1 μL/mL). Color development was performed in the dark at room temperature for 30 min, after which the absorbance of each well was measured at 450 nm. The reaction was stopped by adding 50 μL of 2 N HCl, and the final absorbance was recorded at 492 nm [29]. The maximum absorbance obtained at high concentrations of antivenom serum ($E_{max}$) was interpreted as the maximum binding capacity of the antivenom to the venom antigens [30]. The concentration of the antivenom serum that bound 50% of the venom antigens (corresponding to 50% of $E_{max}$) was calculated as the median effective concentration ($EC_{50}$) [30].

**Chaotropic ELISA.** Chaotropic ELISA was conducted to mitigate the impact of antibodies with weaker antigen binding to snake venom on the ELISA results [29,31]. This procedure followed the standard ELISA protocol, with the key difference being the addition of ammonium thiocyanate ($NH_4SCN$). After the wells were rinsed five times with PBST, 100 μL of $NH_4SCN$ at varying concentrations (0–8 M) was added and allowed to react for 15 min at room temperature. Subsequent steps were the same as those used for the standard ELISA protocol.

### Statistical analysis

Statistical analysis to determine the median lethal dose ($LD_{50}$) and median effective dose ($ED_{50}$) were conducted using the Reed and Muench method [32] and the Probit method [33]. The corresponding median effective ratio ($ER_{50}$) was calculated, representing the amount of venom neutralized per volume unit of antivenom at a 50% survival rate [30]. Using the four parameter logistic curve-fitting in SigmaPlot (version 12.0; SYSTAT Software Inc., CA, USA) [27], the statistical values (mean, standard error, and 95% confidence interval) for $E_{max}$ and $EC_{50}$ were determined.

## Results

### Snake sources and usage

The collection localities, sample numbers (by sex), morphometrics, and venom yields of snakes of each species used in this study are presented in Table 1. A mixed venom sample from three *T. gracilis* samples (collected from Taipingshan) was used for the ELISA. Mixed venom from six *T. gracilis* specimens (collected from Taipingshan and Tataka) was used for $LD_{50}$ and $ED_{50}$ tests. A mixed venom from 12 *T. stejnegeri* (collected from Kaohsiung) was utilized for the $ED_{50}$ assays, as was a mixed venom from seven *P. mucrosquamatus* from the same locality. Additionally, a mixed venom sample from four *D. acutus* individuals (sourced from Southern Taiwan) was used for $ED_{50}$ assays. All snakes were adults.

**Table 1. Comparative morphometrics and venom yields of snake species by collection locality.**

| Species | Locality | Sample size (M: male; F: female) | Snout-vent length (cm) | Tail length (cm) | Body mass (g) | Wet venom yield (mg) | Dry venom yield (mg) |
|---|---|---|---|---|---|---|---|
| *T. gracilis* | Taipingshan | 3 (3F) | 42.6 ± 4.9 (37.5–47.2) | 7.5 ± 2.2 (5.0–9.0) | 48.1 ± 19.1 (29.6–67.7) | 37.62 ± 11.76 (30.56–51.19) | 6.24 ± 1.61 (5.27–8.10) |
| *T. gracilis* | Tataka | 3 (3F) | 43.6 ± 6.3 (36.7–48.9) | 8.9 ± 0.8 (8.0–9.5) | 65.6 ± 15.7 (48.5–79.3) | 49.30 ± 15.52 (33.01–63.92) | 8.03 ± 3.73 (4.87–12.15) |
| *T. stejnegeri* | Kaohsiung | 12 (7M, 5F) | 42.6 ± 4.4 (35.4–49.7) | 9.4 ± 1.5 (7.1–12.0) | 32.7 ± 7.6 (19.0–43.0) | 54.20 ± 19.46 (10.62–160.55) | 12.59 ± 4.60 (2.97–30.82) |
| *P. mucrosquamatus* | Kaohsiung | 7 (3M, 4F) | 74.5 ± 9.5 (57.4–86.1) | 15.8 ± 1.2 (13.6–17.7) | 168.9 ± 64.0 (70.0–252.0) | 110.78 ± 90.02 (47.51–300.21) | 35.02 ± 27.44 (14.59–90.27) |
| *D. acutus* | Southern Taiwan | 4 (3M, 1F) | 84.6 ± 8.4 (78.5–97.0) | 12.3 ± 1.2 (11.0–13.9) | 546.9 ± 173.7 (348.5–758.0) | 179.21 ± 25.38 (147.52–208.46) | 47.07 ± 8.39 (39.45–58.30) |

Data are presented as mean ± standard deviation (range). Venom yield was calculated for each snake on an extraction basis.

## Comparison using SDS-PAGE analyses

To facilitate comparison, we analyzed equal amounts of total venom proteins from *T. gracilis*, *T. stejnegeri*, *P. mucrosquamatus*, and *D. acutus* on the same SDS gel under reducing conditions (Fig 1A) and non-reducing conditions (Fig 1B). Relative band abundances (S1 Table) were determined after scanning each gel lane. The patterns of protein bands differed among the four species under both non-reducing and reducing conditions. However, the precise identification of each band's contents under non-reducing condition was hindered by the absence of proteomic analysis. By referring to the outcomes of proteomic analyses in [15,34,35], the gel under reducing conditions (Fig 1A) revealed the following findings: (1) Diverse banding patterns were observed for snake venom serine proteases (SVSPs) and snake venom metalloproteases (SVMPs) among the four species. Significantly, P-I SVMPs were abundant (20.86%) in *T. gracilis* but scarce in *T. stejnegeri*, and their molecular weights differed from those of *P. mucrosquamatus* or *D. acutus*. (2) Abundant snake venom C-type lectin-like proteins (snaclecs) were present in *D. acutus* (29.75% relative abundance), but were less abundant in *T. stejnegeri*, *T. gracilis*, and *P. mucrosquamatus* (6.33%, 6.94%, and 13.14%, respectively). (3) phospholipase A$_2$s (PLA$_2$s) showed higher abundance in *T. stejnegeri* and *P. mucrosquamatus* (35.13% and 19.27%, respectively) compared to *D. acutus* or *T. gracilis* (1.87% and 9.36%, respectively).

## ELISA

Fig 1 illustrates the immunobinding capacities of the three antivenoms (TsPmAV, GbAV, and DaAV) against the venoms of four pit viper snakes (*T. gracilis*, *T. stejnegeri*, *P. mucrosquamatus*, and *D. acutus*). The optical density of each venom sample stabilized as the antivenom concentration increased. A comparison of the median effective concentration (EC$_{50}$) and maximum binding capacity (E$_{max}$) of the three antivenoms (Table 2) revealed that GbAV showed substantial binding to all four pit viper venoms. TsPmAV showed higher binding to *T. stejnegeri* and *P. mucrosquamatus* venom, but lower binding capacities to *T. gracilis* and *D. acutus* venom. DaAV exhibited better binding only to *D. acutus* venom and poorer binding to the other three pit viper venoms. As a control, non-immunized horse IgG exhibited the weakest binding to all four pit viper venoms, with the optical density rapidly approaching background levels as the concentration decreased (Fig 2).

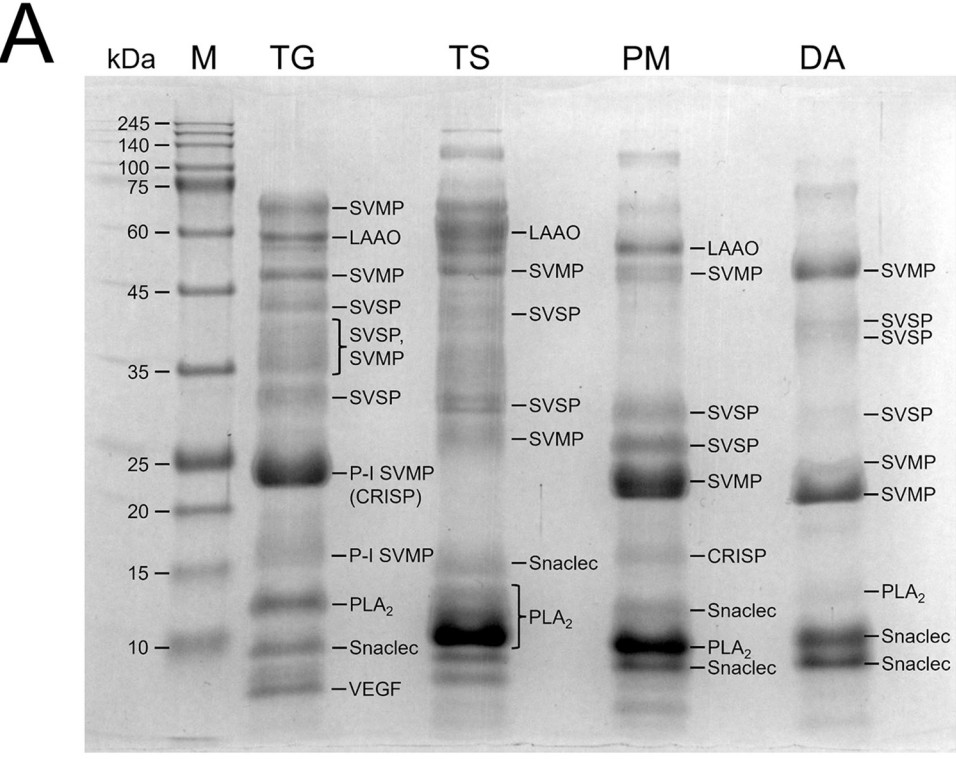

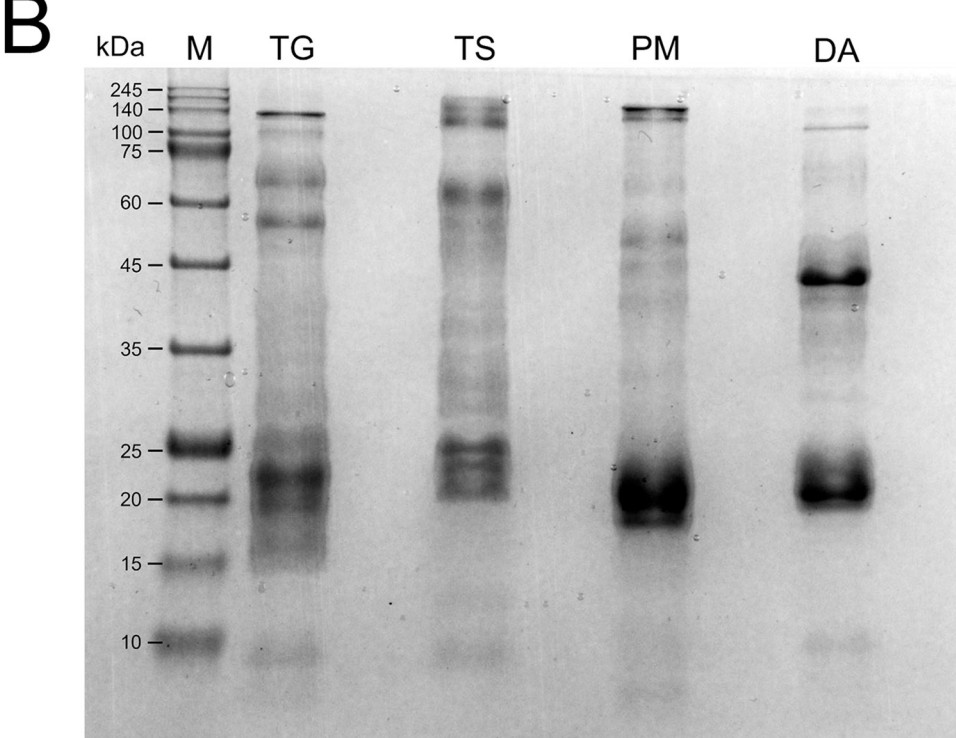

**Fig 1. SDS-PAGE analyses of the venoms of *T. gracilis* (TG), *T. stejnegeri* (TS), *P. mucrosquamatus* (PM), and *D. acutus* (DA), with M representing molecular weight markers.** For each lane, 15 μg of pooled venom proteins were applied, with the proteins being presented in both reduced conditions (A gel) and non-reduced conditions (B gel). The bands in the (A) gel are designated following the results of proteomic analyses in [15,34,35]. Abbreviations: CRISP, cysteine-rich secretory protein; LAAO, L-amino acid oxidase; PLA₂, phospholipase A₂; Snaclec, snake venom C-type lectin like protein; SVMP, snake venom metalloprotease; SVSP, snake venom serine protease; VEGF, vascular epithelial growth factor.

**Table 2. Efficacy of three types of antivenoms for different venom sources.**

| Venom sources | Antivenoms | | | | | |
| --- | --- | --- | --- | --- | --- | --- |
| | *T. stejnegeri* and *P. mucrosquamatus* | | *G. brevicaudus* | | *D. acutus* | |
| | EC$_{50}$ (μg/mL) | E$_{max}$ (OD$_{492}$) | EC$_{50}$ (μg/mL) | E$_{max}$ (OD$_{492}$) | EC$_{50}$ (μg/mL) | E$_{max}$ (OD$_{492}$) |
| *T. gracilis* | 5.96 ± 0.39 (5.15–6.77) | 1.37 ± 0.02 (1.32–1.42) | 1.12 ± 0.14 (0.82–1.42) | 1.48 ± 0.01 (1.45–1.51) | 12.98 ± 0.95 (11.00–14.95) | 1.34 ± 0.03 (1.27–1.41) |
| *T. stejnegeri* | 0.58 ± 0.19 (0.19–0.97) | 2.04 ± 0.03 (1.98–2.10) | 0.24 ± 0.2 (-0.18–0.65) | 1.88 ± 0.04 (1.81–1.95) | 7.31 ± 0.88 (5.47–9.15) | 1.59 ± 0.06 (1.47–1.71) |
| *P. mucrosquamatus* | 0.59 ± 0.16 (0.26–0.93) | 1.71 ± 0.02 (1.67–1.75) | 1.24 ± 0.11 (1.02–1.47) | 1.55 ± 0.01 (1.52–1.57) | 4.17 ± 0.2 (3.76–4.58) | 1.54 ± 0.02 (1.51–1.57) |
| *D. acutus* | 5.28 ± 0.36 (4.54–6.02) | 1.57 ± 0.03 (1.52–1.62) | 1.11 ± 0.14 (0.81–1.41) | 1.68 ± 0.02 (1.64–1.72) | 0.42 ± 0.11 (0.19–0.65) | 1.77 ± 0.01 (1.74–1.80) |

Data determined in triplicate are presented as mean ± standard error (95% confidence interval). Abbreviations: The median effective concentration, EC$_{50}$; maximum binding capacity, E$_{max}$.

## Chaotropic ELISA

We employed the chaotropic ELISA to ascertain the strength of the interaction between antivenom and snake venom antigens. By incubating the antivenom, venom, and chaotropic agents (NH$_4$SCN) together; NH$_4$SCN effectively disrupted the interactions between antivenom proteins and venom antigens. Fig 3 shows the relative affinity differences between the three antivenoms toward *T. stejnegeri* and *T. gracilis* venoms. The binding capacity of all three

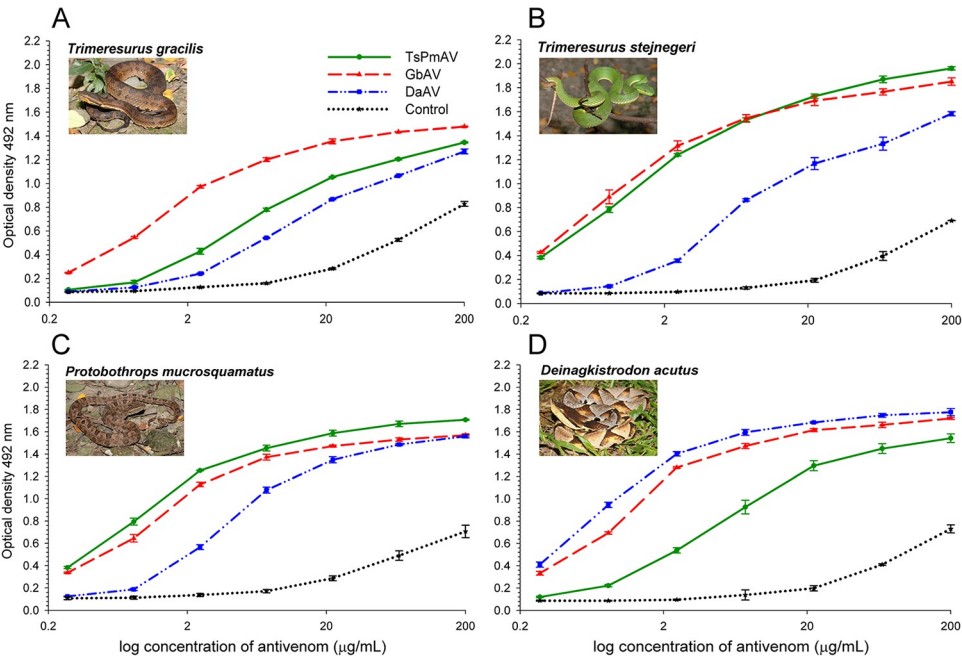

**Fig 2.** Comparative *in vitro* binding reactions of antivenom for *T. stejnegeri* and *P. mucrosquamatus* (TsPmAV), antivenom for *D. acutus* (DaAV), antivenom for *G. brevicaudus* (GbAV), and non-immunized horse serum IgG (Control) toward the venoms of *T. gracilis* (A), *T. stejnegeri* (B), *P. mucrosquamatus* (C), and *D. acutus* (D), using enzyme-linked immunosorbent assay (ELISA). Data determined in triplicate are presented as mean ± standard deviation. The photos in the inset were captured by the corresponding author (TST).

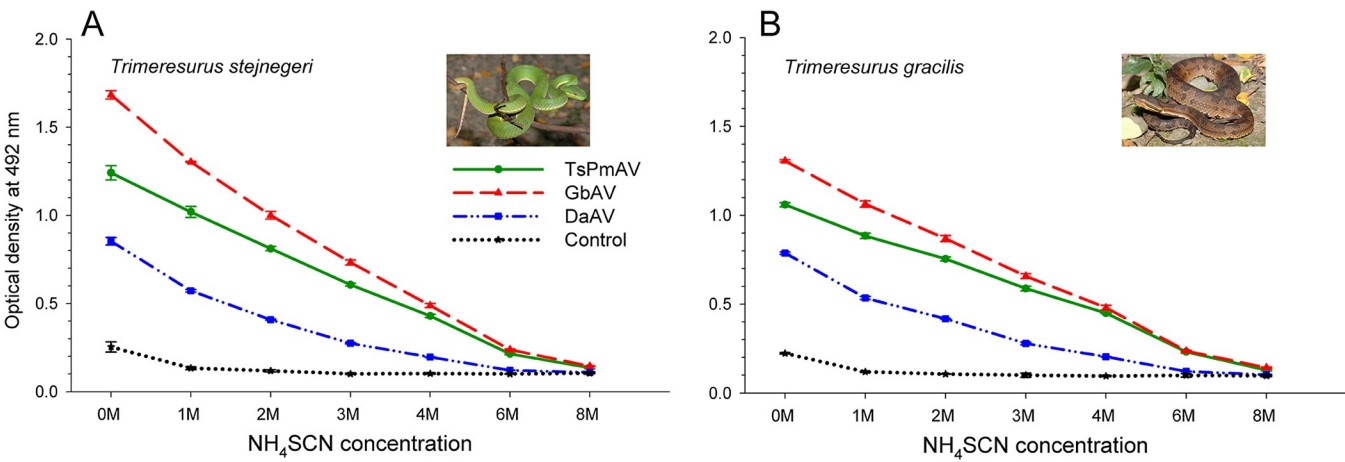

**Fig 3.** Comparative *in vitro* binding reactions of antivenom for *T. stejnegeri* and *P. mucrosquamatus* (TsPmAV), antivenom for *D. acutus* (DaAV), antivenom for *G. brevicaudus* (GbAV), and non-immunized horse serum IgG (Control) toward the venoms of *T. stejnegeri* (A) and *T. gracilis* (B) at varying $NH_4SCN$ concentrations, using chaotropic enzyme-linked immunosorbent assay (chaotropic ELISA). Data determined in triplicate are presented as mean ± standard deviation.

antivenoms to both types of venom decreased with increasing concentrations of the chaotropic agent. The binding capacity ranking of the three antivenoms to both venoms remained largely unchanged regardless of the presence of the chaotropic agent.

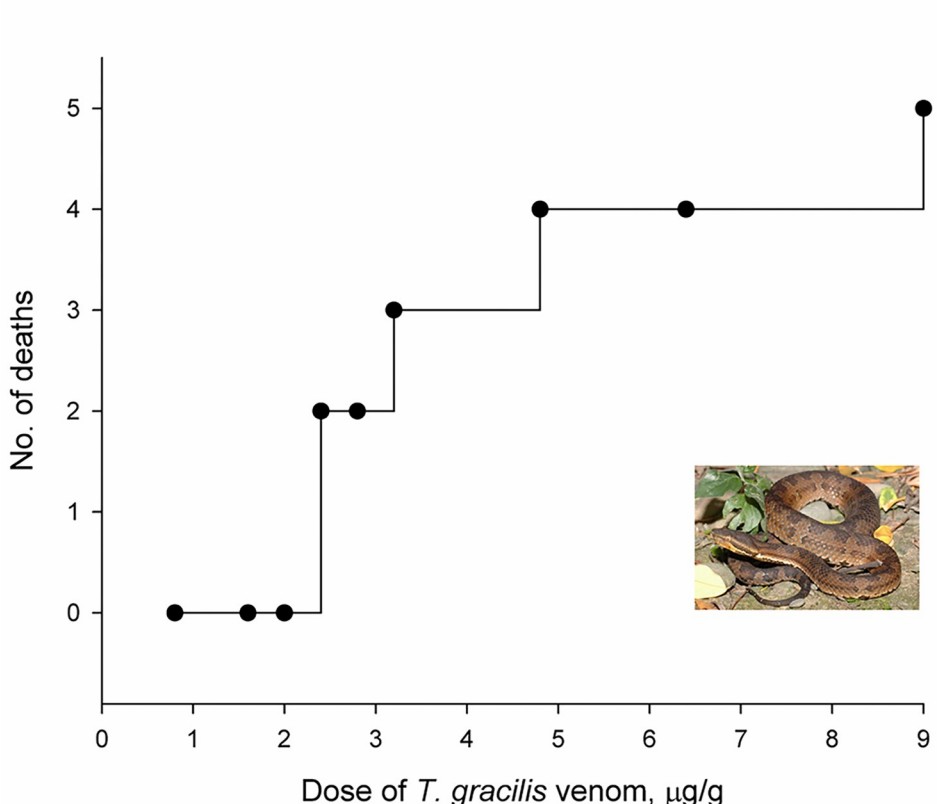

**Fig 4. Mouse mortalities 24 hours after intraperitoneal injection of varying doses of *T. gracilis* venom in nine groups of ICR/CD-1 male mice (five per group).**

### Lethality of snake venom and cross-neutralization by antivenoms

*T. gracilis* venom is lethal to mice (Fig 4), with an $LD_{50}$ calculated at approximately 2.99 μg/g using the Reed and Muench method and approximately 3.02 μg/g using the Probit analysis. In this study, the fixed venom dose for the cross-neutralization assays was set at three times the $LD_{50}$, equating to 153 μg (= 3 × 3 μg/g × 17 g) per mouse, given that the average weight of the mice was 17 g and the $LD_{50}$ was established at 3 μg/g. Combining the fixed amount (153 μg) of venom with two types of antivenoms (TsPmAV and DaAV), it was discovered that TsPmAV more effectively neutralized the lethality of *T. gracilis* venom compared to DaAV (Fig 5A and 5B). The median effective dose ($ED_{50}$) for TsPmAV was approximately 184.62 μL as calculated by the Reed and Muench method and 196.80 μL by Probit analysis. For DaAV, the $ED_{50}$ was approximately 248.88 μL using the Reed and Muench method and 213.56 μL using Probit analysis. When converting the $ED_{50}$ to the corresponding median effective ratio ($ER_{50}$), the ratio of venom to the volume dose of antivenom at which 50% of mice survived, the $ER_{50}$ for TsPmAV (0.78 or 0.83 mg/mL) was higher than that for DaAV (0.61 or 0.72 mg/mL).

## Discussion

In Taiwan, patients envenomated with *T. gracilis* are currently treated with cross-neutralization using TsPmAV because of the lack of serum specific for *T. gracilis* [5]. Accessing to antivenom for *Gloydius* is not easy in Taiwan; therefore, this treatment option is not available to patients. However, TsPmAV has shown limited effectiveness in neutralizing the local effects in patients. Studies have indicated that the sequences of some SVMPs, SVSPs, and $PLA_2$s in *T. gracilis* venom are similar to those found in *Gloydius* and *Crotalus* [21,22]. Therefore, we hypothesized that antivenoms for snake species with venom compositions similar to those of *T. gracilis*, such as *Gloydius* and *Crotalus*, would have a superior immunoreactivity compared to other pit viper species-specific antivenoms already produced in Taiwan, such as those for *T. stejnegeri*, *P. mucrosquamatus*, and *D. acutus*. The experimental results supported this hypothesis.

In this study, we found that the antivenom produced in China for *Gloydius* species (GbAV) exhibited a higher *in vitro* binding capacity or immunological reactivity against the venoms of four Taiwanese viper species (*T. gracilis*, *T. stejnegeri*, *P. mucrosquamatus*, and *D. acutus*)

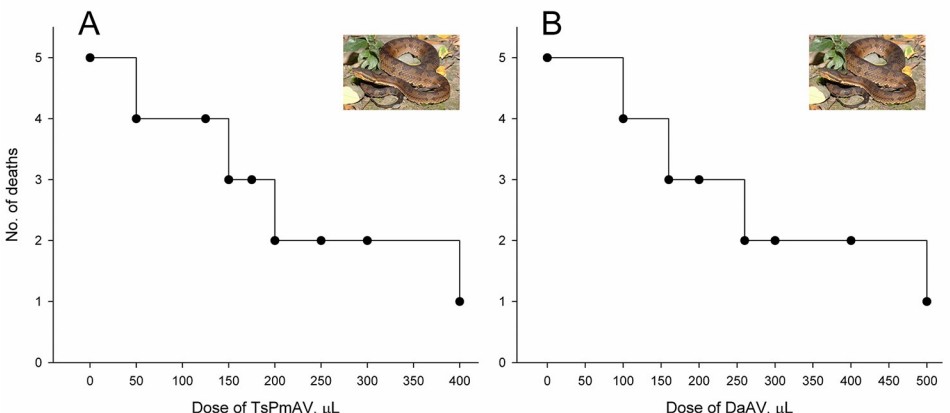

**Fig 5. Mouse mortalities 24 hours after intraperitoneal injection of venoms premixed with antivenoms.** (A) A fixed dose of *T. gracilis* venom (153 μg/mouse) with varying volumes of the antivenom against *T. stejnegeri* and *P. mucrosquamatus* (TsPmAV; 20 mg/mL) in nine groups of ICR/CD-1 male mice (five per group); (B) A fixed dose of *T. gracilis* venom (153 μg/mouse) with varying volumes of the antivenom against *D. acutus* (DaAV; 20 mg/mL) in eight groups of ICR/CD-1 male mice (five per group).

compared to the other two antivenoms (TsPmAV and DaAV), as illustrated in Figs 2 and 3. On the other hand, the *in vivo* toxicity tests conducted on mice indicated that TsPmAV more effectively neutralized the lethality of *T. gracilis* venom than DaAV (Fig 5A and 5B). However, owing to insufficient doses of antivenom for *Gloydius* or *Crotalus*, we could not perform tests to compare their neutralization efficacies against *T. gracilis* venom. We anticipate that the results would align with those of the *in vitro* binding capacity tests, showing a higher degree of toxic cross-neutralization. We look forward to future experiments to obtain sufficient doses of antivenom for *Gloydius* or *Crotalus* [36] to further validate the feasibility of using these antivenoms in the treatment of *T. gracilis* envenomation.

By integrating the SDS-PAGE results depicted in Fig 1, along with the *G. brevicaudus* venom findings in Qin et al. [37], and considering the comparisons of relative venom abundances among different snake species illustrated in Fig 5 of Tse et al. [34], it becomes evident that snaclecs are plentiful in *D. acutus*, but not in *G. brevicaudus*, *T. stejnegeri*, *T. gracilis*, or *P. mucrosquamatus*. The compositions of SVSPs and SVMPs also differ between *T. gracilis* and *D. acutus*. These observations could partially elucidate why DaAV exhibits less immunoactivity against *T. gracilis* venom compared to TsPmAV and GbAV. The abundance and molecular weight of P-I SVMPs in *T. gracilis* venom are unique from other four species. This implies that the toxins present in *T. gracilis* venom may not be entirely recognized by traditional antivenoms such as GbAV, TsPmAV, and DaAV, as demonstrated by the *in vivo* and *in vitro* results presented in this study. In an ongoing study investigating the binding capacity of TsPmAV to *T. gracilis* venom through immunoaffinity column chromatography, we have preliminarily observed lower reactivity with P-I SVMPs and certain SVSPs/PLA$_2$s when exposed to the antivenom.

Commercial antivenoms may exert broad paraspecific immunological binding and neutralization of medically important snake venoms such as in *Bothrops* [38]. Sousa et al. [39] compared the composition and reactivity with multispecific *Bothrops* antivenom (SAB) of venoms collected from six species of *Bothrops*, *Bothropoides* and *Rhinocerophis* snakes, and found that P-III SVMPs are the most antigenic toxins in the venoms of snakes from the *Bothrops* complex, while class P-I SVMPs, SVSPs and PLA$_2$s reacted with SAB in lower levels. Silva et al. [40,41] discovered that the antivenom targeting *Bothrops* venom has limitations in addressing local effects such as hemorrhage and neutralizing SVSPs. However, they observed that supplementing the antivenom with a serine protease inhibitor can enhance its overall efficacy. In subsequent studies, exploring the neutralization efficacy by adding inhibitors specific to P-I SVMPs or SVSPs/PLA$_2$s to the existing antivenom could be considered to enhance therapeutic success for both local and systemic symptoms occurred in *T. gracilis* envenomation.

This study has several limitations: (1) Challenges were encountered in acquiring a sufficient quantity of foreign antivenoms, particularly those targeting *Gloydius* or *Crotalus*. In our upcoming studies, we intend to delve deeper into the compositions of *T. gracilis* venom neutralized by both domestic and foreign antivenoms. This investigation will encompass the utilization of multiple techniques, including western blotting, immunoaffinity chromatography, high-performance liquid chromatography, and mass spectrometry, as well as ELISA and neutralization tests with animals. (2) A rescue experiment, as conducted in [42], which more closely simulates a real envenomation case than the preincubation neutralization experiment, was not carried out in our study. However, it is noteworthy that a majority of relevant studies (such as [30,38,39,41,43–46]) commonly employ the latter design. We opted for preincubation neutralization experiments in our research to facilitate comparisons with other studies. Nonetheless, we recognize the importance of incorporating rescue experiments in future research. (3) We refrained from employing an internal control for normalization across plates that may facilitate a more accurate comparison of E$_{max}$ values. Nevertheless, many published articles on

ELISA (such as [29,30,38,39,43–45]) did not incorporate an internal control. The impact of variation across ELISA plates has been minimized because we implemented measures include maintaining a consistent reaction time and temperature, using identical batch numbers for ELISA plates, employing the same ELISA reader, and ensuring the consistent participation of the same experimenter. (4) As the venom samples from each species are pooled from individual snakes, we refrained from testing differences among treatments using t-tests, ANOVA, or nonparametric statistics, as observed in other studies (such as [29,30,38,42,43,45,46]). For future studies, venom samples from different individuals, populations, or geographical areas can be employed and compared in relevant experiments to enable statistical testing among treatments.

## Conclusion

The bivalent antivenom against *T. stejnegeri* and *P. mucrosquamatus* was partially effective in neutralizing the venom of *T. gracilis*. Both *in vitro* binding capacities and *in vivo* neutralization efficacies were slightly superior to those of the antivenom against *D. acutus*. Additionally, *in vitro* studies have shown that antivenom for *Gloydius* species has a higher immunological binding efficacy against the venom of *T. gracilis* than bivalent antivenom against *T. stejnegeri* and *P. mucrosquamatus*, offering insights into the phylogenetic relationships of *T. gracilis* for biologists. Comparisons of electrophoretic profiles revealed that *T. gracilis* venom has fewer snaclecs than *D. acutus* and more P-I SVMPs or fewer PLA$_2$s than *G. brevicaudus*, *T. stejnegeri*, or *P. mucrosquamatus*, which could partially elucidate the immunoreactivity differences among different antivenoms toward *T. gracilis* venom. This study could provide a reference and substantiation for the treatment of clinical cases involving *T. gracilis* snakebites. Future studies should continue to identify the components of *T. gracilis* venom that are not neutralized by bivalent TsPmAV or other candidate antivenoms, which will be crucial for improving cross-neutralization treatments for envenomation by *T. gracilis*.

## Supporting information

**S1 Table. Relative abundances of labeled protein bands in Fig 1A.** The densitometric intensities of all bands within each venom sample were measured, and the relative abundance of each band was calculated by dividing its intensity to the total intensity of the venom sample. (XLSX)

## Acknowledgments

We would like to thank Dr. Inn-Ho Tsai for inspiring the conceptualization of this study. Additionally, we thank Jun-Wei Chang, Tong-Yu Ke, Mo Han Ruan and Jui-Hsiang Fan for helping in the collection of snake specimens and venom samples. We extend our gratitude to Dr. Hsueh-Ling Cheng and Yu-Han Su for their assistance with the densitometric analysis of SDS-PAGE gels.

## Author Contributions

**Conceptualization:** Po-Chun Chuang, Tein-Shun Tsai.

**Data curation:** Po-Chun Chuang, Tein-Shun Tsai.

**Formal analysis:** Po-Chun Chuang, Jia-Wei Chen, Yuen-Ying Chan, Tein-Shun Tsai.

**Funding acquisition:** Po-Chun Chuang, Tein-Shun Tsai.

**Investigation:** Tein-Shun Tsai.

**Methodology:** Jia-Wei Chen, Yuen-Ying Chan, Tsz-Chun Tse, Tein-Shun Tsai.

**Resources:** Po-Chun Chuang, Yu-Wei Chiang, Tein-Shun Tsai.

**Supervision:** Po-Chun Chuang, Tein-Shun Tsai.

**Visualization:** Jia-Wei Chen, Tein-Shun Tsai.

**Writing – original draft:** Po-Chun Chuang, Tein-Shun Tsai.

**Writing – review & editing:** Po-Chun Chuang, Yu-Wei Chiang, Tein-Shun Tsai.

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
