## [Decision Letter · Decision Letter 0]

12 Feb 2024

Dear Dr Tsai,

Thank you very much for submitting your manuscript "In vitro and in vivo cross-neutralization of Trimeresurus gracilis venom using antivenoms toward four pit viper species" for consideration at PLOS Neglected Tropical Diseases. As with all papers reviewed by the journal, your manuscript was reviewed by members of the editorial board and by three independent reviewers. 

Following the valuable feedback from the reviewers, I kindly request your assistance in making minor revisions to the manuscript. This adjustment will greatly enhance its chances of being reconsidered for publication. I sincerely appreciate your collaboration in this matter.

Sincerely,

Manuela Pucca, Pd.D.

Guest Editor

José María Gutiérrez

Section Editor

Reviewer's Responses to Questions

**Key Review Criteria Required for Acceptance?**

**Methods**

-Are the objectives of the study clearly articulated with a clear testable hypothesis stated?

-Is the study design appropriate to address the stated objectives?

-Is the population clearly described and appropriate for the hypothesis being tested?

-Is the sample size sufficient to ensure adequate power to address the hypothesis being tested?

-Were correct statistical analysis used to support conclusions?

-Are there concerns about ethical or regulatory requirements being met?

Reviewer #1: The methods are justified and clearly described. The level of detail is enough to reproduce the results without havind to consult multiple references. The methods are addecuate to test the presented hypothesis and the results support the conclusions. The well-being of the snakes used in the study is granted and the number of animals used in the in vivo experiment is justified to reach the necessary statistical significance.

The calculation of Emax should be explained further as comparison accross plates in ELISA is difficult without an external control to normalize the absorbance intensity.

Reviewer #2: Are the objectives of the study clearly articulated with a clear testable hypothesis stated?

Yes, the objectives of the study "In vitro and in vivo cross-neutralization of Trimeresurus gracilis venom using antivenoms toward four pit viper species" is articulated with a testable hypothesis. This study contributes to reinforce the importance of detailed and in-depth knowledge of the composition of venoms for the development of effective antivenoms on the treatment of snakebite. In addition, this article shows that in 2024, there are no specific antivenoms for some species of snakes of medical importance. However, the authors with data available in the literature could better discuss about the venoms’ composition, about toxins, such as serine proteases from the venom of snakes of the genus Bothrops, which are known not to be totally neutralized by their specific antivenom, and which are involved in clinical aspects of Bothropic envenomation, such as coagulopathy, as observed by the authors of this article. “However, only two case reports available to date indicate that patients bitten on the finger by this 70 species, not only experience local swelling, bruising, hemorrhagic blisters, or tissue necrosis but 71 also systemic symptoms such as coagulopathy”.

-Is the study design appropriate to address the stated objectives?

Yes, the study design is appropriate to address the objectives. However, experiments, such as an SDS-PAGE of the Trimeresurus gracilis venom compared to the other venoms studied here, could help in the understanding of the differences in compositions. For these reasons, I strongly suggest that the authors do this analysis.

The authors could correlate electrophoresis data and associate with the literature to improve the discussion about the levels of antivenoms binding capacity and knowledge about the toxins that are not recognize by traditional antivenoms. 

Besides this, another experiment, as western blotting could improve the antivenoms binding capacity comprehension. However, the authors mentioned that the problem about the antivenoms quantity.

-Is the population clearly described and appropriate for the hypothesis being tested?

Yes, the population described the hypothesis tested. Characteristics, such as number, sex, size and reproductive status of each snake were described.

-Is the sample size sufficient to ensure adequate power to address the hypothesis being tested?

Considering the difficulties and ethical issues in the use of animals, the sample size is sufficient to ensure adequate to address the hypothesis.

-Were correct statistical analysis used to support conclusions?

Statistical analysis is unclear. Authors should describe better and in more detail the method used and the differences observed.

-Are there concerns about ethical or regulatory requirements being met? 

All animal procedures were approved by the Institutional Animal Care and Use Committee of 218 the National Pingtung University of Science and Technology (approval no. NPUST-110-100, 219 NPUST-112-010)

Reviewer #3: (No Response)

**Results**

-Does the analysis presented match the analysis plan?

-Are the results clearly and completely presented?

-Are the figures (Tables, Images) of sufficient quality for clarity?

Reviewer #1: The results confirm the hypothesis and support the conclusions. The results are clearly presented in a chronologicla manner that allos for the correct interpretation of the results. The tables are of sufficient quality. However, the figures (specially the ELISA results) can benefit of better resolution if the authors want to keep the snake photo on each corner. If that is the case, for consistency, please put the same snake photo in last two figures. Figure footnages are clear and don't require modifications-

Reviewer #2: Does the analysis presented match the analysis plan?

Due to the lack of serum specific for treatment of envenomation by T. gracilis, victims are currently treated using a bivalent antivenom developed against T. stejnegeri and P. mucrosquamatus. This antivenom has limited effectiveness to treat the local effects. So, the authors investigated the cross-neutralization of T. gracilis venom using the bivalent antivenom. The results indicated that the bivalent antivenom was not completely effective in neutralizing T. gracilis venom. 

-Are the results clearly and completely presented?

Yes, the results are clear. However, an electrophoresis analysis is necessary. I also suggest that the next lethality tests be delineated by a way that is closer to the reality of the accident. This could increase the correlation between the binding capacity of antivenoms and the efficacy of protection against the lethality caused by envenomation. 

Also, the term "cross-neutralizing" should be replaced by "binding capacity", since the ELISA assay is an experiment of evaluating binding and not neutralizing activity.

-Are the figures (Tables, Images) of sufficient quality for clarity?

Yes, the figures and images showed good quality.

Reviewer #3: (No Response)

**Conclusions**

-Are the conclusions supported by the data presented?

-Are the limitations of analysis clearly described?

-Do the authors discuss how these data can be helpful to advance our understanding of the topic under study?

-Is public health relevance addressed?

Reviewer #1: The conclusions are supported by th data presented. The only limitation that I miss ackowledgment for is that the preincubation neutrlaisation experiment does not replicate a real envenomation case in which and it would be necessary a rescue experiment in which the venom is injected and then the antivenom is given. However, this is an excellent starting point to determine which antivenom to use that has immediate impact in the clinical treatment of envenomation. Public health relevance is addressed.

Reviewer #2: -Are the conclusions supported by the data presented?

Yes, the data demonstrated that the bivalent antivenom against T. stejnegeri and P. mucrosquamatus is partially effective in neutralizing the venom of T. gracilis.

-Are the limitations of analysis clearly described?

Yes, but the authors should describe better and in more detail the statistical analysis.

-Do the authors discuss how these data can be helpful to advance our understanding of the topic under study?

Yes, next studies should continue to identify the components of T. gracilis venom that are not neutralized by bivalent antivenom, which will be crucial for improving cross-neutralization treatments for envenomation by this snake.

-Is public health relevance addressed?

Yes. This study contributes to reinforce the importance of detailed and in-depth knowledge of the composition of venoms for the development of effective antivenoms on the treatment of snakebites. In addition, this article shows that in 2024, there are no specific antivenoms for some species of snakes of medical importance. These findings provide a reference for developing clinical treatment protocols and contribute to phylogenetic studies.

Reviewer #3: (No Response)

**Editorial and Data Presentation Modifications?**

Reviewer #1: (No Response)

Reviewer #2: (No Response)

Reviewer #3: (No Response)

**Summary and General Comments**

Reviewer #1: The manuscript "In vitro and in vivo cross-neutralization of Trimeresurus gracilis venom using antivenoms toward four pit viper species" compared the use of four antivenoms (1 bivalent and 2 monovalent) available in the Taiwan region for the treatment of Trimeresurus gracilis snakebite. The tests are performed in vitro (ELISA and chaotropic ELISA) and in vivo (preincubation experiment) confirming that the monovalent antivenom raised against Deinagkistrodon acutus is more effective than the bivalent antivenom used nowadays in which one of the species is also a Trimeresus specie.

Minor

 • Line 52: As it is presented right now on the results and discussion, the findings do not offer a clear way to develop more effective antivenoms. I suggest that the authors include in the discussion a note about how this results can guide the development of cross-neutralizing monoclonal antibodies or the development of better cross-neutralizing formulations to keep such statement and improve the reach of the discussion

 • Line 77: Why was the mixture incubated at 37C?

 • Preincubation but not rescue

 • Table 1: T. Stejnegeri Dry venom yield (mg): 4.60 - keep significant ciphers throughout the table

 • The ELISA values can vary a lot depending on the time left for the assay to reveal. Without a internal control that allow to normalize across plates, Emax value cannot be compared across plates. Also, they are close to saturation of absorbance signal (which usually occurs around 2.0 absorbance units).

 • Line 261: T.gracilisvenom (add space between)

Reviewer #2: The article "In vitro and in vivo cross-neutralization of Trimeresurus gracilis venom using antivenoms toward four pit viper species" contributes to reinforce the importance of detailed and in-depth knowledge of the composition of venoms for the development of effective antivenoms. In addition, this article shows that in 2024, there are no specific antivenoms for some species of snakes of medical importance. However, the authors with data available in the literature could better discuss about the venoms’ composition, about toxins, such as serine proteases from the venom of snakes of the genus Bothrops, which are known not to be totally neutralized by their specific antivenom, and which are involved in clinical aspects of Bothropic envenomation, such as coagulopathy, as observed by the authors of this article. Experiments, such as an SDS-PAGE of the Trimeresurus gracilis venom compared to the other venoms studied here, could help in the understanding of the differences in compositions and the authors could correlate electrophoresis data and associate with the literature to improve the discussion about the levels of antivenoms binding capacity and knowledge about the toxins that are not recognize by traditional antivenoms. The bivalent antivenom was not completely effective in neutralizing T. gracilis venom. So, to identify the components of T. gracilis venom that are not neutralized by bivalent antivenom, will be crucial for improving the cross-neutralization and the development of the specific and effective antivenom to treat the victims of this snake.

Therefore, I recommend to accept this article with minor revision.

Reviewer #3: The study proposed by Chuang and colleagues is quite interesting, concise, and objective, demonstrating cross-neutralization of available antivenoms for treating poisonings in Taiwan.

I have some suggestions to enhance the quality of the work:

1. In the study's introduction, the authors could highlight the epidemiology of these envenomings in Taiwan, including the number of cases for each species investigated in the study, mortality rates, sequelae, etc.

2. Present the statistical analysis in the results section.

3. Why were very short dose intervals used in the LD50 assays (0.4-9 ug/g)?

4. The discussion of this study could be improved by providing more data on snakebite accidents and linking them to the neutralization data obtained here.

PLOS authors have the option to publish the peer review history of their article (what does this mean?). If published, this will include your full peer review and any attached files.

Reviewer #1: No

Reviewer #2: No

Reviewer #3: Yes: Isadora Sousa de Oliveira

Figure Files:

Data Requirements:

Reproducibility:

References

---

## [Editor Report · Decision Letter 1]

14 Mar 2024

Dear Dr Tsai,

We are pleased to inform you that your manuscript 'In vitro immunoreactivity and in vivo neutralization of Trimeresurus gracilis venom using antivenoms targeting four pit viper species' has been provisionally accepted for publication in PLOS Neglected Tropical Diseases.

Best regards,

Manuela Pucca, Pd.D.

Guest Editor

José María Gutiérrez

Section Editor

---

## [Editor Report · Acceptance letter]

21 Mar 2024

Dear Professor Tsai,

We are delighted to inform you that your manuscript, "*In vitro* immunoreactivity and in *vivo neutralization* of *Trimeresurus gracilis* venom with antivenoms targeting four pit viper species," has been formally accepted for publication in PLOS Neglected Tropical Diseases.

Best regards,

Shaden Kamhawi

co-Editor-in-Chief

Paul Brindley

co-Editor-in-Chief
